# Monocytic Myeloid Derived Suppressor Cells in Hematological Malignancies

**DOI:** 10.3390/ijms20215459

**Published:** 2019-11-01

**Authors:** Giuseppe Alberto Palumbo, Nunziatina Laura Parrinello, Cesarina Giallongo, Emanuele D’Amico, Aurora Zanghì, Fabrizio Puglisi, Concetta Conticello, Annalisa Chiarenza, Daniele Tibullo, Francesco Di Raimondo, Alessandra Romano

**Affiliations:** 1Division of Hematology, AOU “Policlinico-Vittorio Emanuele”, 95125 Catania, Italy; palumbo.ga@gmail.com (G.A.P.); lauraparrinello@tiscali.it (N.L.P.); cesarinagiallongo@yahoo.it (C.G.); puglisi.fabri@gmail.com (F.P.); ettaconticello@gmail.com (C.C.); annalisa.chiarenza@gmail.com (A.C.); diraimon@unict.it (F.D.R.); 2Department of Clinical and Molecular Biomedicine Ingrassia, University of Catania, 95125 Catania, Italy; emanueledamico82@gmail.com (E.D.); aurora.zanghi@yahoo.it (A.Z.); 3Dipartimento di Chirurgia generale e specialità medico-chirurgiche, CHIRMED, University of Catania, 95125 Catania, Italy; 4BIOMETEC, Dipartimento di Scienze Biomediche e Biotecnologiche, University of Catania, 95125 Catania, Italy

**Keywords:** microenvironment, multiple myeloma, chronic lymphatic leukemia, chronic myeloid leukemia, lymphoma MDSC

## Abstract

In the era of novel agents and immunotherapies in solid and liquid tumors, there is an emerging need to understand the cross-talk between the neoplastic cells, the host immune system, and the microenvironment to mitigate proliferation, survival, migration and resistance to drugs. In the microenvironment of hematological tumors there are cells belonging to the normal bone marrow, extracellular matrix proteins, adhesion molecules, cytokines, and growth factors produced by both stromal cells and neoplastic cells themselves. In this context, myeloid suppressor cells are an emerging sub-population of regulatory myeloid cells at different stages of differentiation involved in cancer progression and chronic inflammation. In this review, monocytic myeloid derived suppressor cells and their potential clinical implications are discussed to give a comprehensive vision of their contribution to lymphoproliferative and myeloid disorders.

## 1. Introduction

With the nomenclature of myeloid derived suppressor cells (MDSCs) we identify a heterogeneous population of immature and mature cells of myeloid origin able to elicit T-cell anergy, promoting tumour immune-escape, via several mechanisms that include depletion of tryptophan, arginine, and cysteine due to the high expression level respectively of 2,3 indoleamine dioxygenase (IDO-1) and arginase (Arg-1), nytrosylation of T-cell receptor, and increased production and release of reactive oxygen species (ROS) [1,2,3]. The complex biochemistry rewiring of MDSCs is due to increased phosphorylation of signal transducer and activator of transcription (STAT) protein family members STAT1, STAT3 and STAT5 [4,5,6,7,8], via decrease of SOCS3 and activation of Janus kinase-signal transducer [9]. In the era of immune-therapies in cancer, there is an increasing interest on how MDSC can re-shape the tumor microenvironment or distant sites, affecting T-cell function and numbers. In this manuscript, we will review the clinical impact of mo-MDSC expansion in hematological tumors, including lymphoma, myeloma, and myeloproliferative neoplasms for their impact on the outcome of patients undergoing immunotherapy.

The amount of Lin−/lowHLA-DR−CD11b+CD33+ MDSCs detected in peripheral blood of normal subjects and patients affected by hematological malignancies is shown in Figure 1.

## 2. Monocytic Myeloid Derived Suppressor Cells in Hematological Malignancies

In tumor-bearing mice, two distinctive CD11b+Gr1+ mononuclear subpopulations are distinguished based on Ly6G expression in monocytic (Ly6G-, low “side-scattered light”- SSC) and polymorphonuclear (Ly6G+, high SSC) tumour-induced MDSCs. In humans, MDSCs subpopulations are termed monocytic (mo-MDSCs, CD14+HLA-DRlow/-) and granulocytic MDSC (g-MDSC, CD33+CD14+HLA-DRlow/-) [2,4], even if phenotype characterization requires a functional assay, like in vitro inhibition of proliferation or IFN-γ production by T-cells [10]. In absence of a human marker equivalent to Ly6C in mice [11], mo-MDSCs can be distinguished from monocytes based on the expression of MHC class II molecule, HLA-DR, while g-MDSCs can be separated from normal neutrophils by the increased expression of lectin-type oxidized LDL receptor 1 (LOX-1) [3]. A third sub-type of cancer-associated fibroblasts has recently identified as CD33+CD13+CD15+IL-4Rα+CD14-CD11chiHLA-DR+ [12,13], called fibrocystic f-MDSCs for their immune-suppressive effect on T-cells, but phenotypically distinguishable from g- and mo-MDSCs [14]. In human cancer patients, current evidence suggests a complex alteration of myeloid cell differentiation and inhibition of their terminal differentiation in polymorphonuclear [15] and monocytic cells [16] in a not completely understood two-step process [4]. First, during the early stages of cancer onset and development, chronic inflammation favors accumulation of immature myeloid at intermediate stages (MDSC-like cells). In the further stages of tumorigenesis, neoplastic cells can attract MDSCs by secreting factors such as granulocyte-macrophage colony stimulating factor (GM-CSF), stem cell factor (SCF), and interferon-γ (IFN-γ). Then, the cytokine milieu, specific for each cancer sub-type and microenvironment, leads MDSCs to complete acquirement of their immune-suppressive properties. For example, inflammatory monocytes and mo-MDSCs migrate and rapidly differentiate toward tumor associated macrophages (TAM) in tumor tissues [6,17], reducing the activity of the transcription factor STAT3 [5]. There are a lot of evidences about the expansion and accumulation of MDSC in the tumour and spleen of tumour-bearing mice, but rarely in lymph nodes [8]. Only one group showed that MDSCs can reduce responsiveness to antigen outside these organ sites by affecting trafficking of T- and B-cells and reducing their expression of CD62L [18], thus exerting a wide-spread, systemic immune suppression in distant lymph-nodes. In our Institution, from 2014 through 2016, we evaluated 375 patients with hematological malignancies (Figure 1); mo-MDSCs were identified as CD45+CD33+CD15-CD14+HLA-DR- in peripheral blood according to our internal procedure previously described [19,20], and we found that in comparison with a pool of 45 healthy subjects, mo-MDSCs were increased in all newly-diagnosed patients tested, except those affected by Waldenstrom disease. The highest percentages of mo-MDSCs was detectable in patients carrying mantle cell lymphoma and chronic lymphatic leukemia (respectively, 52.5 ± 8.1 versus 34.5 ± 2.1%, *p* = 0.02), while patients with Hodgkin lymphoma (HL), follicular lymphoma (FL) and diffuse large B-cell lymphoma (DLBLC) had comparable percentages (respectively 20.3 ± 2.3 vs 29.3 ± 5.3 vs 27.2 ± 3.8%), with widespread values associated to advanced stage. Among plasma cell dyscrasias, multiple myeloma (MM) patients carried higher percentage of mo-MDSCs than those affected by monoclonal gammopathy of uncertain significance (MGUS), respectively, 17.2 ± 1.7 versus 11.1 ± 1.2%, *p* = 0.0003, as we previously disclosed. Among myeloproliferative neoplasms, percentage of mo-MDSCs were low in Philadelphia-negative cases, with no differences between polycythemia vera (PV), essential thrombocythemia (ET) and primary myelofibrosis (PMF) that, polled together, were lower than chronic myeloid leukemia (CML, respectively, 9.3 ± 1.2 versus 19.2 ± 3.2%, *p* = 0.002).

## 3. Monocytic Myeloid Derived Suppressor Cells in Lymphoma

Lymphomas are cancers that originate in the lymphatic system. In Hodgkin Lymphoma (HL) rare neoplastic cells are surrounded by inflammatory and accessory cells, unable of mounting effective anti-tumor immune responses, and that in the last years have emerged as crucial players in sustaining the course of disease [20,21,22,23]. In non-Hodgkin lymphomas (n-NHL) several subtypes are characterized by different cell of origin, metabolism and immune-adaptive response [24]. Due to the success of novel agents and immunotherapy, there is an increasing interest to understand how different immune cells interfere and contribute to the pathogenesis of lymphoma development and immune-evasion [25,26]. For example, targeting MDSCs with histamine dydrochloride enhances in vivo the anti-tumor efficacy of immune-checkpoint inhibitors [27], and a reduction of MDSCs is associated to clinical favorable outcome also in relapsed patients [19,27]. In a pioneer paper in the field, in the A20 B-cell lymphoma model, MDSCs mediated expansion of regulatory T-cells (T-reg) via arginase; and treatment with the arginase inhibitors (N(omega)-hydroxy-nor-l-arginine, NOHA, sildenafil) abrogated T-reg proliferation and tumour-induced tolerance in antigen-specific T cells, disclosing the contribution of MDSCs in lymphoma environment [28,29]. In Eµ-MYC tumor-bearing mice, mo-MDSCs expansion in the lymphoma environment is sustained by soluble mediators, including IDO and arginase. Liposomal doxorubicin treatment could selectively deplete mo-MDSCs in tumor-bearing hosts, as exogenous arginine supplementation partially overcame monocytes’ suppression of T-cell proliferation in vitro, while depletion of granulocytes with anti-Ly6G antibody did not affect tumor growth or host survival [30]. MDSCs are rare in lymph nodes [8], but they can affect lymphocyte trafficking and antigen-induced priming by reducing the expression of homing receptors on naïve T- and B-cells [31,32,33,34] to restrict antigen-driven expansion of both T- and B-cells [18], an effect reverted by gemcitabine-based chemotherapy [31]. MDSCs are increased in several lymphoma subtypes and are associated with a poor clinical outcome, probably due to enhanced chemotherapy resistance as shown by preclinical models [35,36].

## 4. Monocytic Myeloid Derived Suppressor Cells in Hodgkin Lymphoma

In HL, MDSCs count is an independent outcome predictor, comparable to the major widely recognized factor, the interim-2-[18F]Fluoro-2-deoxy-D-glucose Positron Emission Tomography (PET-2) performed in the mid of chemotherapy [37], as shown by two independent single-institution series. Our group showed that three MDSCs subtypes, including CD14+HLA-DR-mo-MDSCs, were increased in newly-diagnosed advanced-stage HL patients and reduced after two and six cycles of ABVD chemotherapy [20]; similarly, in relapsed/refractory patients, several MDSCs subtypes, including mo-MDSCs, were increased at time of disease progression and reduced after treatment with brentuximab vedotin [19]. However, immune-suppressive effect and correlation with outcome is stronger for g-MDSCs than mo-MDSCs in HL [20,38,39]. Another study demonstrated that g-MDSCs were increased in both HL and B-cell NHL patients and correlated significantly with unfavorable prognostic index scores and an inferior outcome [38]. In a third small series, only g-MDSCs (but not mo-MDSCs) were increased in HL patients compared to healthy controls [39].

## 5. Monocytic Myeloid Derived Suppressor Cells in Non-Hodgkin Lymphoma

The mechanisms of tumor-associated immunosuppression in NHL, a known risk factor of progression and poor outcome, is still largely unknown. In follicular lymphoma (FL), diffuse large B cell lymphoma (DLBLC) and mantle cell lymphoma (MCL), an increase of monocyte count at presentation is associated to inferior outcome [40,41,42], reflecting the complex network of by-stander myeloid and neoplastic cells, and recently associated to levels of circulating arginase and IDO [43]. In NHL patients, mo-MDSCs had impaired interferon-α production due to increased STAT1 phosphorylation. Patients with increased ratios of mo-MDSCs to monocytes had more aggressive disease and suppressed immune functions [43]. The defective T-cell proliferation could be restored ex vivo adding arginine.

In FL, gene expression signature associated to expansion of non-malignant tumor-infiltrating immune cells is predictive of clinical outcome [44]. In the pre-Rituximab era, the amount of TAMs was associated to inferior disease free survival and overall survival, a finding recently confirmed for patients treated according to R-CHOP schedule [45], with controversial results based on the kind of treatment offered [46]. However, a comparative evaluation of the clinical impact of MDSCs expansion in peripheral blood is not currently available, and only one group described the expansion of MDSCs in FL by mass cytometry [47]. In diffuse large B-cell lymphoma (DLBLC), mo-MDSCs are increased, associated with T-regs expansion [48], as disclosed in three independent single-center series. Preclinical models disclose that MDSC expansion is associated to chemokine release (e.g., IL-10 [49], CCL3, CCL4, CCL5 [7]) and recruitment of T-regs [7,24,28,49], with the contribution of CD11b+CD27+NK-cells [49]. MDSCs were higher in the non-germinal center subtypes and associated to IPI and R-IPI scores [50,51] and lactate dehydrogenase (LDH) level [49] in the larger series of 144 patients, including 63 GCB and 81 non-GCB DLBCLs [50,51]. In a cohort of 29 newly diagnosed DLBLC patients, mo-MDSCs absolute count (cut-off median count 7.42 × 10^6^/L cells) was associated to inferior event free survival [48]. The same group disclosed that gene expression signature in peripheral blood of DLBLC patients is associated to MDSC expansion and clinical outcome [48]. In lack of data, prospective series are required to establish if mo-MDSCs can be used as a biomarker of outcome in FL, DLBCL and MCL.

## 6. Monocytic Myeloid Derived Suppressor Cells in Chronic Lymphocytic Leukemia

In chronic lymphocytic leukemia (CLL) monoclonal B cells engage in complex, incompletely defined cellular and molecular interactions in the bone marrow, where they initially proliferate, and in the secondary lymph-nodes, where they migrate in the most advanced stages of tumor progression. The so-called nurse-cells in the microenvironment sustain the survival and proliferation of CLL cells, promoting T-cell anergy.

Although studied minimally in CLL, MDSCs play a key role in suppressing T-cell function and proliferation. In 79 consecutive untreated patients affected by chronic lymphocytic leukemia (CLL) mo-MDSCs were increased with suppressing activity against T-cells due to increased indoleamine 2,3-dioxygenase (IDO) activity associated to T-regs expansion, a reduction of chronically activated CD45RA+ effector memory (TEMRA, CCR7− CD45RA+) cells resulting in impaired immune responses. IDO+ myeloid cells could be detected in CLL lymph nodes, but the concentrations of tryptophan and its key degradation products in serum specimens from patients with the highest frequency (MDSChi (*n* = 10)) and the lowest frequency (MDSClo (*n* = 10)) of aberrant monocytes, did not detect any difference, suggesting that the comprehension of clinical impact of mo-MDSCs in CLL is still largely unknown [52].

Relevant for future immunotherapy, tryptophan metabolites produced by increased IDO activity can inhibit CD19-CARTs (Chimeric Antigen Receptor T-cells). While IDO protein is expressed in malignant B-cells of NHL [53], and intratumoral levels are higher in lymphoma than in reactive lymph nodes, associated to poor outcome [54,55,56], in CLL IDO is produced by microenvironment cells, such as dendritic cells, macrophages and nurse-like cells [57]. Fludarabine and cyclophosphamide can downregulate IDO expression in malignant B-cells [58], but there are not data about the off-target effects on mo-MDSCs of Bruton tyrosine kinase (BTK) inhibitors currently used in clinical management of CLL patients. In solid malignancies, BTK is expressed by murine and human MDSCs, whose function can be affected by in vitro and in vivo exposure to the inhibitor Ibrutinib [59].

Gustafson et al. reported that the amount of mo-MDSCs is associated with a shorter time to CLL progression: the six patients with higher mo-MDSCs had a shorter time to disease progression (median 6.9 months) compared to 19.1 months for patients with lower levels (*n* = 22; *p* = 0·024) [60]. In an independent series of 49 CLL patients, upregulation of mo-MDSCs significantly inhibited the CD4+ T-cell immune response and were correlated with the presence of CD4+ T- and CD5+CD19+ cells, contributing to disease progression and a poor prognosis. Indeed, patients with higher MDSC levels experienced an accelerated accumulation of circulating malignant cells while patients with a sustained response to treatment had a decrease in the frequency of mo-MDSC 12 months after completion of treatment compared to measurement prior to treatment [61].

## 7. Monocytic Myeloid Derived Suppressor Cells in Plasma Cell Dyscrasias

In Multiple Myeloma (MM) there is an uncontrolled proliferation of monoclonal plasma cells (PC) within the bone marrow. A monoclonal proliferation of PCs characterizes also other conditions including monoclonal gammopathy of undetermined significance (MGUS) and asymptomatic or smoldering myeloma (SMM), that represent pre-clinical phases of MM. Indeed, virtually all cases of MM pass through a MGUS phase, although it is often not recognized, with low evolution rate in MM ranging 1–10% per year [62,63]. The progression from MGUS to MM is due to both acquired genetic and epigenetic changes of PC associated to an immune suppressive microenvironment re-shaping [64].

In MM, MDSCs contribute to the onset of active disease [65,66,67,68], including angiogenesis [68] sustained by release of metalloproteinase-9 [69], and osteoclastogenesis [67], since MDSC can work as osteoclast precursors [66,67,68]. Moreover, MDSCs promote MM cells growth via secretion of suppressive factors [35,70,71,72,73,74,75,76,77,78,79,80]. MM exosomes can enhance angiogenesis and directly promote endothelial cell growth and modulate signal transducer and activator of transcription 3 (STAT3), c-Jun N-terminal kinase, and p53 in BM stromal cells. Conversely, exosomes derived from bone marrow stromal cells can be taken up by MDSCs, inducing their expansion [73,74] and promoting their survival via triggering of Mcl-1 [80], to further facilitate MM progression. The MM-promoting effect of mo-MDSC is also direct, mediated in part by AMPK activation [81] and still under investigation.

MDSCs expansion is an early event in MM onset as shown in pre-clinical models [79] and MM patients’ bone marrow [78]. T-cell dysfunction is an early event that can be already detected in individuals with MGUS and not fully reverted even when MM patients achieve clinical remission [82].

Stromal cells in the MM bone marrow microenvironment can educate myeloid precursors to differentiate in g-MDSCs [72]. In general, g-MDSCs are more immune-suppressive [35,72,75,77] and have larger tumor promoting effect on MM cells than mo-MDSCs [79], since g-MDSCs can trigger piRNA-823 expression, which then promoted DNA methylation and increased the tumorigenic potential of MM cells [71].

Novel agents active against MM, such as lenalidomide, and bortezomib affect antigen presentation by reducing NF-κB activity [83], in both neoplastic and surrounding cells. Recently, modulation by lenalidomide of regulatory cells, with an immunophenotype overlapping with mo-MDSC, defined as CD33+CD11b+CD14-HLADR-, has been reported [84], while in vitro studies exclude a direct effect of lenalidomide or bortezomib in MDSC expansion [78]. However, in lymphoma-bearing mice, lenalidomide can reduce MDSCs and reverts cancer-induced immunosuppression [85], thus the effect of lenalidomide in MDSC modulation could be tumor-specific.

In 90 relapsed/refractory patients treated with lenalidomide and low dose dexamethasone, mo-MDSC frequency increased without significant correlation with clinical outcome [86]. The levels of mo-MDSC decreased after proteasome inhibitory therapy, as shown in patients and in in vitro experiments [87]. In our experience, g-MDSC were reduced in peripheral blood of patients treated up-front with lenalidomide and increased after bortezomib exposure, without affecting mo-MDSC counts [82].

MDSCs mediate suppression of myeloma-specific T-cell responses through the induction of T-cell anergy and T-reg development. Dissecting the molecular contribution of MDSC subpopulations can be useful for designing and tailoring more effective immunotherapies in MM, especially after the first line of treatment [88] or in the presence of continuative exposure to steroids. Indeed, in non- oncologic settings, MDSCs from glucocorticoid-treated mice increase and can strongly suppress T- cells, dendritic cells, and macrophages, making it more difficult to understand the global picture of immune changes occurring during treatment [89,90].

Standardized detection of MDSC subpopulations and larger series are required to better describe the early changes in immune cell subsets in both newly-diagnosed and refractory MM patients, especially with the increasing access to daratumumab treatment that can target CD38+ immune cells, [91]. CD38 is ubiquitously expressed on neoplastic plasma cells, regardless of the phase of the disease. Daratumumab is an innovative first-in-class biologic targeted to CD38, approved in monotherapy or in combination with novel agents lenalidomide or bortezomib for the treatment of relapsed/refractory patients in Europe and US. Several studies showed that CD38 is expressed on the surface of both PMN and mo-MDSC, suggesting the existence of off-target effects of daratumumab, able to warrant long-term control of disease, by depletion of immune regulatory cells and skewing of T-cell repertoire [91].

In myeloma-bearing mice, decitabine can inhibit neoplastic PC proliferation and induce enhanced autologous T-cell immune response by depleting mo-MDSC in the microenvironment [92]. Some chemotherapeutic agents used also in MM, such as cyclophosphamide or anthracyclines, have immunological side effects [93] that include MDSC expansion [94,95] or inhibition [96] associated to T-cell function modulation, but this aspect has been poorly evaluated in MM. The in vitro cytotoxic effect of melphalan on MM cells can be inhibited in presence of mo-MDSC, and the amount of mo-MDSC before high-doses of melphalan and autologous stem cell transplantation (ASCT) is correlated to poor clinical outcome [97]. However, mo-MDSC count does not affect the clinical outcome of patients undergoing ASCT [97] or allogeneic bone marrow transplantation [98], since the major role in immune suppression in these settings is played by regulatory T-cells. Nitrogen-containing bisphosphonates, such as zoledronic acid, have a predominant role in the supportive therapy of MM patients [99] with beneficial survival [100,101] and delaying tumor growth, due to promotion of MDSC maturation and consequent recovery of T-cell function in murine models of solid tumors [102]. However, how bisphosphonates can affect MDSC count and behavior in MM patients has never been explored.

## 8. Monocytic Myeloid Derived Suppressor Cells in Myeloproliferative Neoplasms

Chronic myeloid leukemia (CML) is a myeloproliferative neoplasm due to the engagement of an aberrant onco-protein BCR-ABL. The targeted therapy based on tyrosine kinase inhibitors (TKIs) is the current standard of care for patients in chronic phase, leading to long-term control of disease due to the achievement of deep molecular responses. In selected patients, TKI therapy can be discontinued [103]. However, TKI discontinuation can cause rapid disease relapse, presumably due to the restart of dormant neoplastic stem cells in permissive environment, a very active research field. Immune escape mechanisms predominate within the CML microenvironment in newly diagnosed patients, associated to increased amount of MDSCs and T-regs [104]. Our group disclosed that CML cells promote MDSC expansion through the release of soluble factors and exosomes, creating an immune-tolerant environment that results in T-cell anergy and favors tumor growth [105,106].

In contrast to other hematological malignancies [107], there is an important overlap between tumor cells and MDSC in CML, because both g-MDSC and mo-MDSC express, at least in part, the driver onco-protein BCR/ABL protein [106,107,108]. Imatinib and dasatinib treatment decrease the number of MDSCs and their biochemical biomarkers IL-10, Arg1 and MPO [109]. Our group previously showed that mo-MDSC subset was efficiently decreased only in patients who received dasatinib, and not imatinib or nilotinib [106], disclosing an effect independent from the suppression of clonal myelopoiesis, despite the number of persistent mo-MDSCs correlated with the major molecular response after exposure to dasatinib [106]. In accordance with these data, Hughes and colleagues showed that patients in MMR and MR4.5 have increased NK cells and restoration of effector function, consistent with reduced numbers of mo-MDSCs and reduced T cell PD-1 expression [104]. In this emerging scenario, dynamic evaluation of immune changes occurring during treatment with TKI could contribute to better define the deep molecular response and may promote treatment-free remission.

Myelofibrosis (MF) is a chronic Philadelphia-negative (Ph-) hematological neoplasm characterized by hematopoietic stem cell-derived clonal myeloid proliferation, featured by an inflammatory condition driving the progression of disease. Bone marrow fibrosis is the result of a complex and not yet fully understood interaction among megakaryocytes, myeloid cells, fibroblasts, and endothelial cells [110] triggered by auto-immune phenomena [111]. The oncogenic over-activation of JAK2 signaling can lead to the endothelial-to-mesenchymal transition in the microvessels close to megakaryocytes [112], and maintains high PI3K signaling over the threshold required for CXCR4 activation favoring increased trafficking of hematopoietic stem cells from the bone marrow microenvironment to extramedullary sites [113]. The pathological emperipolesis of neutrophils within megakaryocytes, leads to the leakage of alpha-granular contents into the bone marrow microenvironment, including the release of PDGF and TGF-beta [110,111], responsible for chronic inflammation onset. In primary myelofibrosis, there is an abnormal activity of key cells of the immune system, including increase in monocyte/macrophage compartment, altered regulatory T-cell frequency, expansion of myeloid-derived suppressor cells, and CD4/natural killer cell dysfunction [114]. Both mo-MDSCs and PMN-MDSC levels are significantly elevated in MPNs compared with controls, without any difference between primary and secondary MPNs, neither correlated with JAK2 status, white blood cells, hemoglobin levels, platelet counts, splenomegaly, or the degree of bone marrow fibrosis [115].

## 9. Conclusions

In all hematological malignancies mo-MDSCs are increased, despite the consequences on clinical outcome being still under investigation. Several strategies to target MDSCs could improve immune therapies via multiple mechanisms:1)MDSC function: phosphodiesterase inhibitors, nitroaspirins, synthetic triterpenoids, COX2 inhibitors, ARG1 inhibitors, anti-glycan antibodies, and IL-17 inhibitors can affect the immune-suppressive and restore T-cell activity;2)MDSC maturation: ATRA, vitamins A or D3 or IL-12 can promote differentiation of g-MDSCs in neutrophils; while N-Bisphosphonates, modulators of tyrosine kinases, and STAT3 inhibitors can affect the transcriptional program of all sub-types of MDSCs;3)MDSC depletion: several conventional chemotherapeutic agents (e.g., gemcitabine, fludarabine and cyclophosphamide) or novel immune agents (e.g., daratumumab) can affect viability of MDSCs and exert off-target effects, relevant in a different way for each specific tumor sub-type.

## Figures and Tables

**Figure 1 ijms-20-05459-f001:**
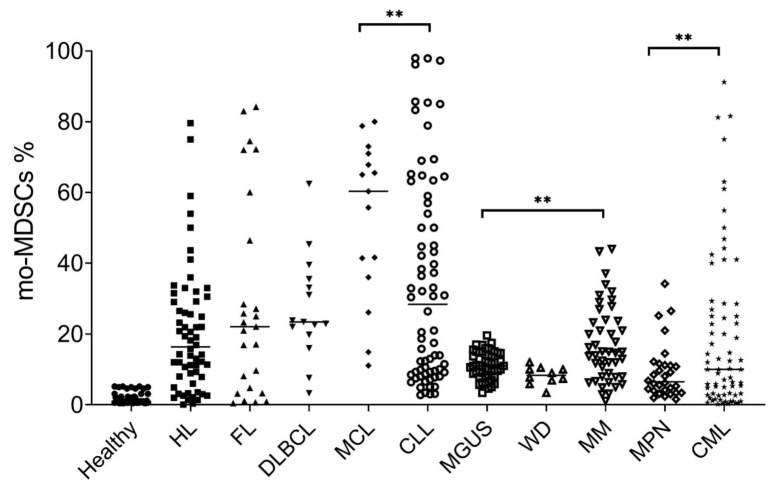
Comparison of peripheral circulating monocytic myeloid derived suppressor cells (mo-MDSCs) proportions in normal controls and patients affected by hematological malignancies (** *p* < 0.001, as indicated; description in the main text).

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
