# Peer review of "Monocytic Myeloid Derived Suppressor Cells in Hematological Malignancies"

_ijms, 2019, doi:10.3390/ijms20215459_

Round 1
Reviewer 1 Report
This is a nice comprehensive review of monocytic myeloid derived suppressor cells in hematologic malignancies.
The article in general is well structured. However, moderate English edits are necessary. In particular, editing of run on sentences would be particularly useful.
Author Response
Dear reviewer 1,
thanks for your comments; accordingly, we edited the manuscript to improve run-on sentences. Changes have been highlighted in yellow in the revised version of the manuscript.
Best regards,
Alessandra Romano
Reviewer 2 Report
This review article describes the roles of monocytic myeloid derived suppressor cells in hematological malignancies. This paper is well organized and will provide several insights into the understanding of immune state in hematological malignancies. Several modifications will make this paper more beneficial to the readers of International Journal of Molecular Sciences. The specific points are as follow.
Specific points
The paragraph 2, “Figure “, is not appropriate as an individual paragraph with regard to constitution. Paragraph 4 contains the description of Hodgkin lymphoma, non-Hodgkin lymphoma, and CLL and this constitution is not appropriate. To combine paragraph 3 and the part of Hodgkin lymphoma and non-Hodgkin lymphoma in paragraph 4 should be considered. The authors described that mo-MDSCs is abundant in CLL patients as compared with lymphoma patients. In addition, they describe that IDO plays an important role for increased mo-MDSCs. The difference of IDO between lymphoma and CLL patients should be described. The effect of daratumumab on mo-MDSCs in myeloma patients should be described.Author Response
Dear reviewer 2,
thanks for your comments; accordingly, we edited the individual paragraphs of the manuscript, and added statements about IDO in lymphoma and CLL and more details about anti-CD38 treatment and MM.Changes have been highlighted in yellow in the revised version of the manuscript.
Best regards,
Alessandra Romano
Round 2
Reviewer 2 Report
The authors adequately modified the manuscript according to the reviewer’s comments.